# “It’s about What You’ve Assigned to the Salad”: Focus Group Discussions on the Relationship between Food and Mood

**DOI:** 10.3390/ijerph20021476

**Published:** 2023-01-13

**Authors:** Megan F. Lee, Joanne F. Bradbury, Jacqui Yoxall, Sally Sargeant

**Affiliations:** 1Faculty of Society and Design, Bond University, Gold Coast, QLD 4217, Australia; 2Faculty of Health, Southern Cross University, Gold Coast, QLD 4225, Australia; 3Faculty of Health, Southern Cross University, Lismore, NSW 2480, Australia

**Keywords:** depression, mental health, psychological wellbeing, qualitative research, focus group, thematic template analysis

## Abstract

Current observational and interventional studies in nutritional psychiatry suggest that healthy dietary patterns rich in fresh whole foods could protect against depressive symptoms, and that unhealthy dietary patterns high in ultra-processed and refined foods could contribute to depressive symptoms. However, no studies have explored detailed subjective accounts behind the food and mood relationship. This study aimed to uncover unknown factors in the human experience with food and mood. Using a phenomenological framework, this focus group study applied thematic template analysis to accounts of over 50 Australians aged between 18 and 72. Three themes were identified from the transcript of the focus groups: (i) reactive and proactive relationships with food, (ii) acknowledgement of individual diversity relating to eating and mental health, and (iii) improving mood by removing food restriction and eating intuitively. The data highlights the complexity of the relationship between food and mood that extends beyond biological mechanisms which could be used to extend current epidemiological and intervention studies in the field of dietary patterns and depression.

## 1. Introduction

Nutritional psychiatry is an emerging field in mental health [1]. The importance that nutrition plays in chronic lifestyle diseases such as cardiovascular disease and type 2 diabetes is generally well understood [2,3], and now several clinical trials suggest that whole-of-diet interventions could be beneficial for treating mental health disorders such as depression [4,5,6,7]. Current observational studies also suggest that healthy dietary patterns rich in fresh whole foods could protect against depressive symptoms, and that unhealthy dietary patterns high in ultra-processed and refined foods could contribute to depressive symptoms [8]. Such findings provide evidence that nutrition could be a complementary treatment for psychopharmacology and psychotherapy for people experiencing mood disorders [9,10]. However, meta-analytic scrutiny has so far yielded inconsistent findings due to methodological issues such as unexplained heterogeneity and risk of bias [11].

Due to these inconsistent findings, it is important to seek more nuanced information regarding the interplay of diet and depression, which may be achieved by exploring individuals’ experiences with food and mood. Experiences of depression and depressive symptoms have been extensively reported within qualitative contexts [12,13,14], yet individual perceptions of food and its relationship to mood remain largely unexplored.

Some published qualitative papers focus on food intake and mood but explore consumer perspectives of functional foods—those specifically designed to prevent chronic disease [15]. Others focus on how consumer mood influences food product choices [16], rather than focusing specifically on individuals’ experiences with diet and mental health. Results from such studies detail how consumer mood is linked to emotional eating and present a potential commercial need for foods that have an anti-depressant influence. Consumers described the ease of eating healthily when in a good mood. In contrast, they described comfort foods as unhealthy and linked to reward systems from childhood [15]. Similarly, food preference and mood were found to be contextually different in different consumers when making product choices [16].

It has been well documented that individuals have a socially constructed idea of how food, and diet influence mental health [17,18,19]. Individuals’ dietary patterns are influenced by life experiences, current beliefs and expectations, social interactions, and the context in which food is selected [20]. These personal contexts, beliefs, and experiences are difficult to measure in a positivist framework and therefore, a deeper phenomenological examination of the perceived relationship between food and mood is necessary. However, no studies have explored detailed subjective accounts behind the food and mood relationship.

Phenomenology is the philosophical enquiry of the structure of human experience and consciousness; it details how individuals’ experience a specific phenomenon from their perspective, context and interpretation [21]. A phenomenological researcher aims to put aside their own experiences, preconceived ideas and biases about a phenomenon and embrace how the phenomenon appears to the participant [22]. The phenomenological inquiry in this study will act as a method and vehicle to explore food and mood related experiences [23].

The extent to which individuals assign meaning to food and the impact on mood needs further research. A phenomenological perspective may assist such examination in a broader context, considering strong social representations about certain foods exist and have done so for many years [24]. Considering that food choice is multifaceted and complex, this study was designed to elicit the relationship between food and mood in more detail than that currently articulated. The exploration of meanings attached to healthy and unhealthy eating, how different eating patterns make people feel, and how they shape their food choices [20], is a logical next step in learning how associations between dietary patterns and depression are formulated.

## 2. Materials and Methods

This qualitative focus group study explores individuals’ experiences with food and mood using the consolidated criteria for reporting qualitative studies, the (COREQ) 32-item checklist [25] can be found in the Appendix A.

### 2.1. Participants and Procedure

Nine focus groups, each of approximately 60 min duration, were conducted in regional Australian locations. The regional locations have a wide range of ethnic and cultural communities. Focus groups were conducted in seven face-to-face and two online formats (via Zoom). All students and staff at a multi-campus university in Australia received an invitation email to participate. Additionally, participation invitations were shared on social media platforms Facebook, Twitter, Instagram, and LinkedIn. Individuals interested in participating were asked to provide their telephone number and a preferred time to call to discuss the study’s details. A follow-up email was then sent which contained an information sheet that outlined the study and a consent form. The study protocol was approved by Southern Cross University Research Ethics Committee #19-083.

### 2.2. Data Collection

Eighty-three emails were received from interested individuals; 50 were able to attend the arranged timeslots for one of the nine focus groups. No compensation apart from refreshments was offered for participation. The focus groups were facilitated by the lead researcher (ML), a female PhD candidate and psychology tutor at Southern Cross University. The first focus group was also attended by the primary PhD supervisor (Assoc Prof SS) as part of the qualitative training within the PhD program, while the following eight were attended solely by ML and the participants. All focus groups were audio-recorded and transcribed verbatim. Field notes were taken directly after each focus group discussion and a reflexive journal was kept, including thoughts, issues, concerns, feelings attitudes, beliefs, values, expectations, ideas, experiences, and potential biases to account for why certain elements were identified during data collection [26,27].

Before each focus group, the facilitator established rapport, explained the personal interests and potential biases that she brought to the research, and described that the purpose of the focus groups was to explore individuals’ experiences with food and mood. The facilitator then prompted group discussion, using questions such as ‘Can you describe how something you ate resulted in changing the way you felt?’, and ‘Can you describe how your feelings and emotions have influenced your eating?’. Participants were not given the list of questions prior, in order to the focus group discussion. A full list of semi-structured questions and prompts used in the focus groups can be found in the Appendix A. Focus group discussions continued until data reached saturation [28].

Of the 50 participants who attended the discussions, 36 were female (72%), and 14 were male (28%). Further demographic information was not collected to ensure minimal intrusiveness of the research and potentially increased information sharing during the discussions [20]). Some (but not all) participants disclosed their age during the focus group discussions; these ranged between 18 and 72 years. It was noted that all participants were either currently undertaking or already held a university degree.

### 2.3. Data Analysis

A six-stage thematic template analysis was used to explore the data [29]. During stage one, data were transferred from the audio recording tool to Otter software (https://otter.ai/ (accessed on 12 September 2019)) to upload audio recordings into text. Transcriptions were then read line by line against the audio, and corrections made to the text by the lead researcher. All participants were de-identified at this stage in the transcription. An email was sent offering participants the opportunity to read and return feedback on the transcripts. One participant reviewed the transcripts and returned no comments. The remainder of participants were happy not to review the transcripts. Each transcript was then read in its entirety by the lead researcher. During the initial reading, patterns in the text and initial participant quotes of interest were noted. The lead researcher continued to keep account of any thoughts, issues, concerns, feelings attitudes, beliefs, values, expectations, ideas, experiences, and potential biases that could influence data analysis in the reflexive journal.

During stage two, an iterative and inductive coding process was conducted. It began with open coding, in which all transcriptions were reread line by line. Codes—a short word or phrase assigned to label a portion of the data [30])—were organised using NVIVO software, version 12 (QSR International Pty Ltd., Burlington, MA, USA, 2018). Upon completing stage two, themes were identified as patterns across the data which were unified by a central concept and underpinned by the research question [31]. Relationships between codes and themes were discussed, and an initial coding template was constructed. Once the coding template was complete, it was reapplied line by line to each focus group transcript using NVIVO software [32]. Table 1 shows the thematic template analysis table. Upon completion, each theme’s scope and content was defined; some themes were refined into sub-themes. After this stage, working titles were given to all themes, a final coding template was developed, and a thematic map constructed of the included themes.

## 3. Results

An inductive thematic template analysis revealed three themes, (i) reactive and proactive relationships with food, (ii) acknowledgement of individual diversity relating to eating and mental health, and (iii) improving mood by removing food restriction and eating intuitively. These themes were found across all focus groups and included descriptions of participants’ relationships with food, the meaning they attached to food choices, the social context in which food choices were made and the overall relationship between mood and mental health.

### 3.1. Theme One: Reactive and Proactive Relationships with Food

This theme explores participants’ knowledge of how different ways of eating influenced their mood. The theme has two subthemes: (i) values assigned to food, and (ii) agency and direction of choice.

#### 3.1.1. Values Assigned to Food

In this subtheme, participants explored the values that they assigned to food and how this influenced their mood. Discussions were not limited to mood as a result of food consumption or intention to shift mood. On several occasions, participants suggested that their mood was influenced by values assigned to the food before it is eaten. Josephine describes how the values she assigns to the foods she eats directly influence her mood:


*It’s about what you’ve assigned to the salad. If you’ve assigned to the salad, this is healthy, and I’m doing this because I haven’t done this much, then it’s a good thing. However, if I’ve assigned to salad, I’m doing this because I’m restricting, then a salad is the worst thing like for me it will ruin my mood.*
(Josephine)

Josephine suggests that if she believes that she is eating a salad because it is good for her health and what she wants to eat, this enhances her mood. However, if she eats a salad because she is restricting food to lose weight and eating it under duress, this worsens her mood. Josephine describes that fluctuations in mood are less about the food components and more about the value ascribed to the things eaten, possibly inferring that the biological mechanisms of food are secondary to the psychological value placed on what she eats.

Negative values assigned to food were reported across the focus groups. One participant suggests a fear of food learned from childhood. In this extract, Mary describes the fear of food she developed as a young ballerina:


*Well, I think possibly, I am terrified of food from the early ballet years. I’m never just going to eat carelessly unless I’m madly drunk. One time I did cry eating some chocolate mousse at a big meeting in _____, a long, long, long time ago. Other than that, it’s a decision driven occupation. Sometimes it’s good. Sometimes it’s great. Sometimes it’s not quite as good as you thought it might be. The creme brulee that you had to have. I guess. Terrified. I think we’re coming from a position of terror. However, then perhaps in an extremely different way., I’m just not going to put it in my mouth If I’m going to gain weight.*
(Mary)

Strict food restrictions imposed by parents and trainers lead to an unhealthy relationship with food; Mary uses the word ‘terrified’ more than once in her extract to emphasise this. Mary explains that she continues to restrict food in the same way that she was forced to when she was a child, and believes this is entrenched in her adult behaviour. Her description of these behaviours as an ‘occupation’ further highlights the seriousness of the value she assigns to food and its influence on her mood. Another factor is that she strongly controls her food intake unless she is under the influence of alcohol. She suggests that drinking alcohol may help her loosen her food inhibitions. Mary explains that the terror surrounding her relationship with food is deeply entrenched in her fear of gaining weight, which seems to be a side-effect of childhood eating habits. Elizabeth shares a similar view to Mary regarding a negative relationship with food:


*I went on an alkalising diet last year, and it was very strict. I lost way too much weight really quickly. It was too strict. However, you find, food is company, and when you cut it out, you have no comfort anymore. So, all these emotions come out…. I think that’s why things like chocolate and cakes and all the yummy things are quite comforting. I know through years of sort of struggling with weight, that eating is to push the emotions down. If you remove all of that, the emotions are coming out.*
(Elizabeth)

Elizabeth considers a recent diet she tried, and due to the strictness of the food restriction, she had adverse symptoms such as excessive weight loss. Elizabeth suggests the diet impacted her mood because she finds comfort and solace in the foods she enjoys. When she could no longer have these foods, she was deprived of the comfort and company that food would normally provide for her. Elizabeth then acknowledges that this deprivation would increase her negative emotions. Like Mary, Elizabeth also describes that the guilt she experiences with food and her food relationship relates to a societal thin ideal and that she must always be trying to lose weight, which lowers her mood.

#### 3.1.2. Agency and Direction of Choice

Participants described how low mood could make them turn to unhealthy food options, making their mood worse. The interconnection between mood, mental health and food intake was commonly discussed across the focus groups. For example, Amy describes how she feels when eating refined and processed grains, and how this influences her mood:


*If I eat bread in any form, whether it be organic or mass-produced grain that has no nutrition, then I start to get feelings of lethargy, and I start to lose the impulse to have anything energetic happen. I will perpetuate more and more reasons to not get up and do anything in a hurry. It will make me feel like I’ve been less accomplished. Therefore, contributing to more feelings about low worth because I haven’t achieved anything. If I compound the wheat intake with dairy (and I love ice cream), then I know that that also helps get me into a downward spiral of low energy, lethargy, lack of enthusiasm towards the things I need to do, which makes me feel guilt. We totally are affected by what we put into our bodies.*
(Amy)

A vicious cycle is reported between eating what is considered unhealthy food, lack of motivation, energy, lowered mood and feelings of insignificance and lowered self-esteem. Amy describes being stuck in a negative cycle and how difficult it is to stop once it begins. However, Amy also uses the word ‘accomplished’ when describing food intake, indicating that she may link food to a sense of achievement. Although she only describes the negative cycle of unhealthy food intake, it suggests that she may also be aware of what foods to eat that help maintain a positive mood and lead to feeling more accomplished.

Much of the discussion about reactive relationships with food focused on reward, comfort, or commiseration. Many participants commented that no matter what their mood, reasons always existed to use food to either elevate mood or comfort negative mood. Jesse describes food as a constant reward or commiseration:


*It’s like an excuse in my brain. You’ve done really well. You should go and treat yourself to something nice. Or you’re upset. I should eat something. Whatever mood, I have an excuse to eat, and I use food as my emotional crutch.*
(Jesse)

When Jesse does well, she rewards herself with food. Conversely, when she is upset, she turns to food for comfort and solace. This implies that there is always time to use food to celebrate or commiserate. She acknowledges needing to give herself a reason or an excuse to use food as a reward or comfort. The relationship between food and mood is not always a conscious choice to elevate mood. Food may also be used to wallow, grieve, commiserate, or when bored, despite the health implications of eating that food.

Sugary and convenient snack foods were commonly associated with lowered mood. However, while participants acknowledged this, they disclosed eating them frequently. This cognitive dissonance is acknowledged by Alexa, who describes how her way of dealing with low mood and swings in emotion is by eating more chocolate than she considers wise:


*There would be times where I would hit like an emotional low, and my way of coping with that was going and buying a block of chocolate and just eating it. Like the whole thing. However, then I feel guilt and shame for eating more than I should, and this makes me feel worse than I was feeling before.*
(Alexa)

Alexa suggests that she overindulges when feeling emotional by eating a whole chocolate block rather than a square or row. The consequence of this is that she is not choosing healthy foods to improve her mood but making an unhealthy choice that negatively influences her mood. Despite her knowing that her pattern of chocolate consumption lowers her mood, she seems unable to control her intake, which may be a negative coping mechanism for dealing with her emotions.

Discussions of the bidirectional association between food and mood were common among the participants. Most suggested that their food choices influenced their mood, but their mood also influenced food choices. On several occasions, participants suggested that feeling lazy or tired made them turn to unhealthier, convenient options which they knew would lower their mood.

Participants in each focus group described that the values assigned to the foods they consumed contributed to their mood and overall mental health. They articulated how these values signified rewards, comforts or commiserations, alongside reasons to treat themselves with foods of lower nutritional value. The discussions also included aspects of the role of food guilt. Assigning the value that a portion of food was detrimental to health led to guilt, blame and shame for eating these foods, which in turn was described as lowering mood.

### 3.2. Theme Two: Acknowledgement of Individual Diversity Relating to Eating and Mental Health

In theme two, participants suggested that every person is different physiologically and psychologically, which influences food and mood experiences. Participants discussed how there should not be one uniform set of guidelines for food and mood recommendations. Matilda suggests that each person’s food and mood experiences differ:


*It’s very individualistic. Everybody’s body processes are different. Some people react to one food in one way, and other people react to that same food in a different way. So, what is considered to be healthy, and what’s considered to be unhealthy may not be the same for every single person.*
(Matilda)

Matilda wrestles with the idea that health may mean different things to different people because of the various ways that food impacts people physically and mentally.

In the following extract, Su describes how her individual reactions to food contribute to guilt due to not eating the way deemed healthy or acceptable by society:


*I think it’s also hard not to beat yourself up if you’re doing something that is not working. Like, what is the best? Is it the food pyramid? Is it Paleo? Is it primal? Is it no grains? Not beating yourself up that you’re eating something different than everyone else. Recognising that everyone has their own types of food that works best for their body. I think everyone needs to come to that conclusion. It’s not a one shop fits all for everyone. To me, it’s just confusion, even though I know sort of what works best for me. Sometimes I eat well. Sometimes I don’t. You just do the best you can and not trying to realise you’ve got to understand it all.*
(Su)

Su acknowledges that individual differences in food consumption and mood are common in society. However, Su explains she does not feel guilty about relinquishing a style of eating that does not work for her. The implication of this is that the acceptance of societal diversity enables contentment with her individual choices. She illustrates a catalogue of different media messages regarding changing dietary advice, fad diets, and government guidelines, but finds it important to recognise that food choice and what works for each person are different. She suggests it is difficult to determine what foods to choose on an individual level because uniform guidelines for everyone may not suit one person.

The perception that uniform guidelines are not suitable for whole populations of people of different sizes, shapes, genders, physical activity levels and food tolerances was commonly addressed. Emma agrees with Su that the current population-based dietary guidelines in Australia may not be useful advice for everyone but are the default from most health professionals:


*I know when I’ve been like to see a dietitian in the past, and you’re going there because you’re wanting to get some healthy assistance. It is very much the dietary guidelines and diet pyramid. In the end, you just stop going because this is not what I want. Even with doctors in a way you feel like you aren’t being heard. I don’t want the grains, I don’t want the dairy, I’m looking at an alternative, you know, healthier. It’s frustrating, sometimes you go to the doctor about feeling sad. They don’t know about nutrition, they offer you a pill, you try and do something a little bit different. Go see a dietitian, it’s still that food pyramid and it seems that everyone’s still very behind the eight ball with all of this.*
(Emma)

Emma describes frustrations with health professionals who always provide advice in line with dietary guidelines. Emma describes not feeling heard by health professionals in that she does not want the same advice that has not worked for her in the past, and ultimately, she chooses not to engage with health services because she feels their nutrition advice is outdated and not supported by evidence or a lack of individualised nutrition advice. Emma implies that medication may not be the only option for the treatment of mental ill-health symptoms, and that diet and nutrition could be a lifestyle factor that may be used as an alternative to current treatments for low mood.

Throughout the focus groups, participants described how individuals have different eating patterns and consume different foods. They also suggested that each person’s biological system is different and that foods may react differently for each person. Descriptions of food intolerances and individual dietary needs were discussed, which presented specific physiological and psychological reactions.

### 3.3. Theme Three: Improving Mood by Removing Food Restriction and Eating Intuitively

Participants reflected on the influence of food restriction and dieting compared to eating intuitively and the influence that such consumption patterns had on mood. In this theme, participants discussed intuitive eating and its association with mood—Intuitive eating is defined as an eating pattern that removes food restriction, weight focused thinking and promotes listening to internal hunger and satiety cues, accepting the body’s natural size and shape, eating mindfully, removing food guilt and ending food preoccupation [33]. Participants’ discussed dieting, disordered eating, food preoccupation, body weight and shape, and specifically addressed how these factors negatively impacted mood and mental health.

The detrimental impacts of dieting and food restriction were commonly acknowledged across the focus groups. In the following extract, Shae describes her struggle in the past with food restriction:


*I had orthorexia for a while, without realising it. Because I wanted to be healthy, and then, you know, like, no that food’s good, that food’s bad…….However, I was really anxious and panicky around food. I was protective around food, if I’m cooking, don’t come and pick at it when I’m eating, because that’s less food for me, this may be my only chance to eat. It did change my mood and my mindset, and I still am struggling with myself to try and eat intuitively.*
(Shae)

Shae describes her experience with orthorexia nervosa. Orthorexia nervosa is defined as a disordered obsession with eating only foods that are considered to be healthy. People who experience symptoms of orthorexia avoid any foods they deem as unhealthy, which can create disordered eating patterns over the long term [34]. Shae describes focusing too closely on food and eating healthy foods with little to no dietary flexibility, to the point of it interfering with her mood and mental health. An implication of this is that she denigrates food types, which causes anxiety, panic, and stress. She suggests that the symptoms of orthorexia developed without intention, and that her focus on healthy eating had slowly become uncontrollable. Shae also suggests a territorialism with food triggered by food preoccupation, and a scarcity mindset caused by food restriction. Shae comments that she is still trying to eat intuitively and listen to her body rather than always eating what she deems healthy. Later, Shae suggests that eating the foods that she likes and not restricting foods elevates her mood:

As the mother of a newborn, Katie describes the urgency to lose the weight she gained during pregnancy and how food restriction lowered her mood:


*I went through postnatal depression. I had a huge period that I didn’t realise was disordered eating, like very disordered. Going on the diet bandwagon, lose the baby weight while having postnatal depression. I went on that food restriction, and I felt good losing weight. Then, it’s not sustainable. So, then the binge happens; and then I’m already low mood with the depression, and dealing with the baby, and then adding to it, because food was the coping mechanism or the control factor, I suppose. And then that body image stuff came into play.*
(Katie)

The consequences of restrictive food behaviours during the onset of postnatal depression contributed to Katie’s negative eating patterns and exacerbated her symptoms. She explains that despite this, she continued dieting throughout this time which she believed helped her lose weight, but the weight loss was only for a short time and her deprivation mindset ultimately led to bingeing on the foods she loves. This cycle of dieting and bingeing resulted in an even lower mood, exacerbating her already compromised psychological well-being and the added pressures of parental caretaking responsibilities.

The cycle of dieting and restricting foods is also described by Jane, who suggests that she finds it very hard to stop such a pattern:


*Because without having a good relationship with food and seeing food as like the enemy, or bad or feel guilty about it. You feel bad about yourself because you’re gaining weight. It can really affect my mental health and then the whole dieting cycle begins, which I think is really a big part of like the whole food and mood thing as well. However, when I eat the foods that I like, that my body tells me are good for me, and that I want without restriction—intuitively—when I’m not on a diet that makes me feel good.*
(Jane)

Jane describes the impacts of the cycle of dieting and food restriction as a contributing factor to mood and mental health levels. She explains that viewing particular foods as restricted make her feel more negatively about herself. She implicates food guilt with weight gain and body image problems leading to lowered mood and negative mental health. She also describes that when she is not restricting foods and eating intuitively her mood and mental health improves.

Intuitive eating was a term that was discussed across the focus groups. Participants suggested that dieting and restricting foods was detrimental to positive mood and increased negative mental health. They described eating the foods they enjoyed without guilt and listening to their bodies about which foods made them feel good. This intuitive way of eating was described as elevating mood for many of the focus group participants. They suggested that removing weight-focused thinking and focusing instead on the health foods brought to their body was uplifting for their mood, and that focusing on food guilt, losing weight, and dieting worsened their mood.

## 4. Conclusions

A recurring theme across the focus groups was reactive and proactive relationships with food. Within this theme, many participants suggested a bidirectional link between food choices and mood. It was acknowledged that food choices influenced mood, but mood also influenced food choices. These findings suggest that reverse causality could be a factor in the food and mood relationship. For example, ultra-processed, refined and sugary foods are correlated with lowered mood and increased mood disorder symptoms [5] (but it is also suggested that people who experience low mood and symptoms of depression may be more likely to eat ultra-processed, refined and sugary foods and less fresh natural foods [35,36]; additionally, healthy, whole foods are more expensive and less accessible for people experiencing low mood [37]. The reverse causality of the diet and depression relationship has been well documented [38,39,40,41]. One study found that those who currently experienced depressive symptoms had lower diet quality scores [39]), while others found that lower diet quality was associated with higher rates of depressive symptoms [41] These findings aligned with discussions by the focus group participants who reported that they were more likely to resort to poor food choices when their mood was compromised. They also acknowledged that making poor food choices led to lowered mood and suggested that mood played a role in their food choices. They described tendencies to turn to unhealthy food when sad, tired or stressed, leading to emotional eating; “a tendency to eat in response to negative emotions and particularly for foods high in fat and sugar” [42]. Emotional eating occurs when one cannot distinguish hunger from boredom, irritation, frustration, sadness, anxiety, depression, distraction, or fatigue [43,44]. It has been proposed that the brain signals a bidirectional pathway that drives appetite and a preference for hyper sweet, hyper salty, nutrient-poor foods [45]. This tendency towards emotional eating is most critical in high-stress times, as suggested throughout the focus group discussions [46].

In the subtheme of *values assigned to food,* participants described that although the nutritional components of the food they ate were important for their mood, their relationships with, and the values they assigned to food were of greater importance than the nutrients consumed in the diet. These findings suggest that “there is more to food than just nutrition” [47]. There are many reasons to eat apart from sustenance: boredom, tiredness, distraction, stress, frustration, anxiety, depression or grief [43,44], and foods commonly eaten for these reasons are usually high in sugar and salt, are ultra-processed and provide psycho-physiological effects of instant gratification. Similarly, a positive mood can trigger an increase in appetite and targeting indulgent foods commonly used to reward and celebrate [47]. Research suggests that moods impact people’s liking for different foods; people eat more unhealthily when experiencing low moods and more healthily when experiencing heightened moods [48]. Mood has also been cited as influencing personal appetite, desire to eat and food choices; healthy food choices play a specific role in enhancing mood [38].

Participants also perceived that individual differences determined the efficacy of recommended dietary patterns. This finding in the theme of *acknowledgement of individual diversity relating to eating and mental health* suggests that participants did not support a uniform approach to food and mood guidelines and health promotion. These findings support epigenetic studies that emphasise flexible and integrative approaches to nutrition, with recent recommendations for the future direction of nutrition advice to be personalised to diet type, genes and phenotype [49]. Recent epidemiological studies suggest that food addiction and eating behaviours could be influenced by an individual’s phenotype [50]. It is postulated that addictive eating behaviour is expressed in different psychobiological vulnerabilities to depression, emotional dysregulation, anxiety, low emotional awareness, childhood trauma history, difficulties listening to hunger and satiety cues, and gene abnormalities. The findings from such studies and the current focus group study support Marx, et al. [51]’s assertion that the causes of, influences on, and interventions for mood disorders are complex and not fully understood. This complexity may be why anti-depressant medications and psychotherapies work for some, but not all people who experience the symptoms of depression [52].

Participants also reported frustrations of not knowing which foods were good or bad for physical and mental health due to being surrounded by many different media messages, advertising, and population-based guidelines, such as the national dietary guidelines. This aligns with previous research exploring individuals’ knowledge of dietary guidelines, sources of nutrition information, and factors influencing food choices [53], which found that despite dietary guideline awareness, participants rarely adhered to recommendations and were unsure about serving sizes. The main source of nutrition information found by Lambert, Chivers and Farringdon [53] was social media, which was highly weight-focused rather than health-focused.

In the final theme of *Improving mood by removing food restriction and eating intuitively,* there was a consensus among the groups that intuitive eating and food restriction were important elements in the food and mood relationship. Intuitive eating is a non-diet approach to eating known to positively influence body image, self-esteem and psychological well-being and reduce disordered eating behaviours [54]. Intuitive eating involves removing food restriction and weight-focused thinking, listening to the body’s internal hunger and fullness cues, eating mindfully, accepting body shape and size, removing food guilt, and eating what nourishes the mind and body [55]. The focus group findings are consistent with studies that found that intuitive eating results in reduced depressive symptoms, greater self-esteem, higher body image satisfaction, and fewer disordered eating behaviours [54,55,56]. Other studies report that non-diet approaches reduce disordered eating behaviours and increase psychological well-being and self-esteem [57,58,59]. The focus group data clearly align with this research, implying that eating intuitively and removing food restriction may play an important role in the food and mood relationship outside of the foods or nutrients consumed in an individual’s dietary pattern. These findings suggest that an individual’s relationship with food and the role of psychology in food choices could be integrated into nutritional interventions for mood disorders alongside current treatment options.

This is the first study that has explored individuals’ experiences with food and mood in depth, specifically focusing on mental health. This is also the only paper that has applied a phenomenological lens to individual food and mood experiences and socially constructed representational contexts. There were also limitations of the study that should be addressed. Most of the participants were observed to be health or diet conscious. The participants’ dietary needs were acknowledged before the focus groups took place when asked if there were any dietary restrictions for the provided refreshments. Most participants discussed health-related dietary restrictions such as vegetarianism, veganism, paleolithic or ketogenic diets, or that they adhered to gluten or dairy-free dietary patterns. Future focus group research could ensure the voices of participants who were not as diet-conscious are included. Other limitations common to focus group studies may also have influenced the findings [60].

Overall, the data from the focus groups point to the diversity of experience related by participants in their consideration of the bidirectional relationship between food intake and mood. More specifically, the data emphasise the need for more detailed phenomenological and social representational research in this area. The prevalence of depression is widespread, and the importance of good mental health maintenance is now greatly emphasised. If dietary patterns are to be promoted to influence and improve mental health, research must extend beyond a positivist paradigm. Despite growing statistical evidence supporting the role of nutrition in mental health, high levels of heterogeneity raise more questions relating to subjective perceptions of the links between nutrition and mental health. The demand for more experiential inquiry to fully explore the nuanced relationships between food and mood must continue.

## Figures and Tables

**Table 1 ijerph-20-01476-t001:** Thematic template analysis table.

Nodes	Codes	Themes
Mood influencing food choices	Bidirectional Link Between Food and Mood	Reactive and Proactive Relationships with Food
Food choices influencing mood	Relationships with Food	
Disordered relationships with food	Food as Control	
Comfort food used for low mood	Mind/Body, Gut/Brain Connection	
Foods used as reward or commiseration	Short-Term vs. Long-Term Effects of Food on Mood	
Each person physiologically reacts to food differently.	Bio-individuality	Acknowledgement of Individual Diversity Relating to Eating and Mental Health
Each person psychologically has differing moods	Individual Differences	
The relationship between food and mood is different for everyone	Intolerances	
Dieting and food restriction decreases mood	Food Restriction and Dieting	Improving Mood by Removing Food Restriction and Eating Intuitively
Weight focused thinking	Weight Focused Thinking	
Listening to your bodies cues when it is hungry and when it is full	Listening to Bodies Internal Hunger and Satiety Cues	
Eating mindfully, rather than when distracted	Mindful Eating	
Food guilt causing low mood	Removing Food Guilt	

## Data Availability

The transcripts for this focus group project will be made available by a request to the authors.

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
