# Peer review of "“It’s about What You’ve Assigned to the Salad”: Focus Group Discussions on the Relationship between Food and Mood"

_ijerph, 2023, doi:10.3390/ijerph20021476_

Round 1
Reviewer 1 Report
The authors present an interesting qualitative study exploring the relationship between food and mood. Themes highlighted the bidirectional relationship between food and mood, individual differences and how dieting and restriction impacts mental health. More detail is needed in the methods including following the consolidated criteria for reporting qualitative research (COREQ) guidelines. My comments are below:
Introduction:
1. The introduction provides a good review of the current literature.
2. Some editing is needed to improve the flow.
Method:
3. Details about the research team involved should be reported for qual studies, this includes who lead focus groups, what their credential are, occupation at the time of the study, and importantly any characteristics that may bias interpretation or influence the responses of participants.
4. It is also good practice to follow the COREQ reporting guidelines for qualitative research. I suggest following the guidelines and reporting any additional information missed in the methods. You can also include the COREQ checklist in the supplementary materials, similar to how you include a PRISMA checklist in a systematic review. https://academic.oup.com/intqhc/article/19/6/349/1791966
Results:
5. Description of the results following the quotes are too similar to the original quotes. This feels repetitive. I suggest tying in the interpretation of multiple quotes together to provide more of a description of the theme and not so much detail about direct quotes.
6. Think about including a figure representing the themes as this would provide a visual summary of the results.
Discussion/conclusion:
7. Overall, the discussion does a good job tying in the qualitative findings with wider literature.
8. Most of the quotes seem to be about acute experiences of food and mood. This should be o discussed.
9. Many of the experiences are around restriction/diet culture and how that impacts mood. It would be interesting to compare findings with any literature on disordered eating and mental health.
Author Response
Thank you for taking the time to review our manuscript. Please find attached point by point response to the comments.

Reviewer 2 Report
for mi it was pleasure reading this article. I have few questions rather than objections. title is a little bit confusing and long, i think that authors wanted it to be catchy and intriguing, but i needed to read it three times until i finally understood the meaning.
Also, as a key words, i am not sure that "food and mood" should be key word as a syntax. For me it would be more descriptive food as a comfort, key word qualitative is missing study or research, it cant stand alone.
Line 68, i would never say meaning of food is under-researched; in past decade there has been explosion of research on all levels, of psychological value of food and impact on population both mental and psychological health.
Author Response

(The authors gave the same response as above.)
